# Why Do Nasolabial Folds Appear? Exploring the Anatomical Perspectives and the Role of Thread-Based Interventions

**DOI:** 10.3390/diagnostics14070716

**Published:** 2024-03-28

**Authors:** Gi-Woong Hong, Sehyun Song, Soo Yeon Park, Sang-Bong Lee, Jovian Wan, Kyung-Seok Hu, Kyu-Ho Yi

**Affiliations:** 1Samskin Plastic Surgery Clinic, Seoul 06577, Republic of Korea; cosmetic21@hanmail.net; 2Division in Anatomy and Developmental Biology, Department of Oral Biology, Human Identification Research Institute, BK21 FOUR Project, Yonsei University College of Dentistry, 50-1 Yonsei-ro, Seodaemun-gu, Seoul 03722, Republic of Korea; bokcool21@gmail.com; 3Made-Young Plastic Surgery Clinic, Seoul 06615, Republic of Korea; sy-jumsim@hanmail.net; 4Pygmalion Clinic, Seoul, Republic of Korea; lovechannel99@hanmail.net); 5Asia Pacific Aesthetic Academy, Hong Kong, China; jovian.wan@apaa.org; 6Maylin Clinic (Apgujeong), Seoul 06001, Republic of Korea

**Keywords:** nasolabial fold, barbed threads, reverse technique, cross technique, volumizing thread, cogged thread

## Abstract

The classification of nasolabial folds into three types, each with distinct causative factors and mechanisms, is explored. Age-related changes in facial skin and connective tissues are examined in detail, revealing variations across different facial regions due to variances in tissue firmness and thickness. The innovative ‘Reverse Technique,’ involving cog threads to enhance tissue traction and effectiveness in thread-lifting procedures, is introduced. Detailed technical guidelines, anatomical considerations, and safety measures are provided, emphasizing the importance of identifying optimal vectors and fixing points to achieve maximum lifting effects while minimizing potential risks, particularly those associated with vascular structures. Additionally, the ‘Cross Technique using volumizing thread’ is discussed, designed to smooth tissue boundaries and rejuvenate sagging areas. Facial anatomy, including the positioning of arteries and ligaments, is underscored as essential for ensuring the safety and efficacy of procedures. In conclusion, this review stands as a comprehensive guide for practitioners, offering insights into innovative thread-lifting methods and their applications in addressing nasolabial folds. The primary focus is on achieving optimal aesthetic results while prioritizing patient safety.

## 1. Introduction

The nasolabial folds refer to the pair of skin creases extending bilaterally from the nasal wings to the corners of the mouth. These creases are demarcated by facial features that provide support to the buccal fat pad, establishing a distinctive division between the cheeks and the upper lip [1]. The nomenclature derives from the Latin words “nasus”, denoting the nose, and “labium”, signifying the lip. It is pertinent to acknowledge that the accurate anatomical term for this fold is actually melolabial fold, representing the crease between the cheek and the lip. Over the course of natural aging, these folds may exhibit an increase in both length and depth [2].

In clinical settings, dermal fillers are frequently utilized for their minimally invasive nature, lower likelihood of side effects, and shorter recovery periods [3]. However, the injection of fillers into nasolabial folds poses a potential risk of vascular issues, with severe instances leading to irreversible skin tissue necrosis or permanent vision loss, resulting in blindness [3]. These complications arise from potential damage to the facial artery, which courses along and in proximity to the nasolabial fold [4].

In response to such concerns, facial threads containing cogged and volumizing threads have emerged as an alternative approach to effectively address deep nasolabial folds while minimizing the risk of inadvertent intravascular injections. Examining age-related changes in facial skin and connective tissues reveals variations across different areas based on tissue firmness and thickness. Regions around the head and ears, characterized by minimal subcutaneous fat but robust and thick superficial musculo-aponeurotic system (SMAS) layers, resist sagging due to strong attachment of internal structures within the skin. These regions serve as fixing points, enabling threads to pull and gather sagging tissues during thread-lifting procedures.

Conversely, the skin and connective tissues around the nose, cheeks, and jawline possess thinner SMAS layers and abundant fat tissues. These areas lack strong internal tissue adherence, resulting in more noticeable sagging as individuals age. Ligamentous tissues responsible for linking the skin, SMAS layer, and connective tissues in these areas lose resilience with age, contributing to sagging skin and fat tissues [5,6]. Nevertheless, these ligamentous tissues, composed of fibrous materials tougher than skin and fat tissues, serve as hanging points during thread-lifting procedures, maximizing the pulling and anchoring strength of the threads [7,8,9].

Before introducing a novel thread-lifting method to enhance sagging tissues and wrinkles around the nose and perioral area, an examination of the mechanisms and manifestation of nasolabial folds is essential for a comprehensive understanding of causative factors, guiding the selection of appropriate therapeutic strategies for optimal effectiveness.

## 2. The Etiology of Nasolabial Folds

The nasolabial folds are characterized by lines resembling the Chinese character ‘八’, extending diagonally from the upper border of the nasolabial groove, situated between the nose and the cheek, toward both sides of the cheek. While colloquially referred to as nasolabial folds, categorizing them into three distinct types based on their underlying mechanisms provides valuable insights for targeted treatment selection (Figure 1).

Volume Deficiency in Adjacent Region: This cause is associated with a lack of volume in the area next to the nasal alar. It results in limited superficial fat beneath the nasolabial fold due to a recession of the canine fossa in the maxilla. This can lead to a hollow appearance in the buccal area and the formation of deep lines resembling wrinkles. Historical interventions for this issue included implants or fat grafting, but contemporary practice primarily uses fillers for volumetric correction due to their enhanced elasticity.

Skin Attachment and Sagging: Another cause involves the attachment of the skin below and above the wrinkle line and the upper lip region. Skin below the wrinkle line is firmly attached to the underlying orbicularis oris muscle, resulting in diminished sagging. In contrast, the skin and tissues above the line are loosely connected, leading to sagging of the anterior buccal fat pad and a noticeable bulging appearance. With age, there is often an enlargement of the superficial fat above the wrinkle line and a reduction of the buccal fat below, creating a marked thickness contrast between the tissues above and below the wrinkle line.

Specific Fibers of Upper Lip Elevators: The third cause is primarily attributed to specific fibers of upper lip elevators, such as levator labii superioris alaeque nasi, levator labii superioris, and zygomaticus major and minor muscles.

Therefore, the nasolabial fold should be treated according to the types of the nasolabial fold causes.

The first type is associated with a volume deficiency in the region adjacent to the nasal alar. Individuals with a recession of the canine fossa in the maxilla, a structure supporting the area beneath the nasal alar, typically display limited superficial fat beneath the nasolabial fold, in contrast to the prominent nasolabial fat above the fold. Historical interventions involved implants or fat grafting; however, contemporary practice predominantly employs volumetric correction using fillers due to their enhanced elasticity (Figure 2) [10,11,12,13].

Moreover, in the absence of fat in the deeper layers, the resultant volume deficiency leads to the buccal area appearing hollow in a triangular shape. This gives rise to a step-like layering at the demarcation between the recessed area and the protruding fat layer above the nasolabial fold, resulting in the formation of deep lines that resemble wrinkles, although they are not actual wrinkles. Addressing wrinkles arising from this case involves filling the recessed buccal area. While conventional procedures, such as implants or fat grafting, were previously commonplace for this purpose, contemporary practice predominantly employs volumetric correction using fillers, driven by the enhanced elasticity of modern filler formulations.

The second type pertains to the skin beneath the wrinkle line and the upper lip region. The skin below the wrinkle line exhibits relatively firm attachment to the underlying orbicularis oris muscle, contributing to diminished sagging. Conversely, the skin and tissues above the wrinkle line are loosely connected, resulting in the anterior buccal fat pad sagging downward with age, giving rise to a discernible bulging appearance [1].

With aging, the superficial fat above the wrinkle line, referred to as nasolabial fat, typically undergoes enlargement, while the buccal fat below the wrinkle line, in proximity to the nasolabial fold, tends to diminish. Consequently, a marked thickness contrast emerges between the tissues situated above and below the wrinkle line. Furthermore, the waning elasticity in the skin and tissues above the prominent wrinkle line contributes to their sagging, forming a configuration that overlies the firmly anchored buccal fat, thereby intensifying the depth of the nasolabial fold.

The third type is primarily attributed not to volume disparities among specific tissue areas or sagging caused by aging but rather to specific fibers of upper lip elevators, such as levator labii superioris alaeque nasi, levator labii superioris, and zygomaticus major and minor muscles. These fibers attach to the skin near the nasolabial fold beside the nose before connecting to the upper lip and orbicularis oris. The muscle fibers attached to the nasolabial fold predominantly exert traction on the skin beneath the lower boundary of the nasolabial fat along the fold [14]. Consequently, during smiling, the musculocutaneous attachment points on the skin are pulled, resulting in the concavity of the area below the fold and an increased prominence of the fat layer above the fold, leading to a deepening of the wrinkle line. While in youth, these lines may be prominently visible only during smiling, with age and repetitive movements leaving skin imprints, these lines become more perceptible even without facial expressions (Figure 3).

The most common patterns are a combination of types two and three of the above.

The muscles involved in elevating the lips originate from the zygomatic bone area but descend, gradually becoming superficial as they integrate with the SMAS near the nasolabial fold, traversing along the same layer after merging with the skin. Consequently, when the threads’ protuberances affect the deeper tissues, including the SMAS, they may impede the movement of the integrated SMAS and muscle fibers attached to the skin [15]. Hence, during nasolabial fold thread lifting, deliberate efforts should be made to primarily exert traction on the bulging superficial fat layer above the SMAS, mitigating hindrances caused by the movement of muscles in proximity to the SMAS (Figure 4) [15,16,17].

At this stage, the placement of threads too superficially, close to the skin, poses a potential risk of inducing a folded appearance, akin to dimples, particularly evident when smiling or expressing facial expressions post-procedure. Consequently, it is advisable to ensure a sufficient depth of thread insertion into the fat layer to preempt such occurrences. While each of the three types of nasolabial folds may manifest independently, they typically arise from overlapping causes involving two to three types. In the context of the second type, where threads are employed to exert traction on tissues, such as loose skin and fat above the nasolabial fold line, the previously elevated area above the fold can be leveled, resulting in a reduction of fold depth. This intervention naturally mitigates the relatively sunken appearance near the nasal side of the cheeks.

For the third type, characterized by an increase in fold depth due to muscle fiber attachment to upper lip elevating muscles, interventions involving the introduction of volumizing agents along or across the fold line can produce a volumizing effect to ameliorate this condition [18,19]. The volumizing agent procedure contributes to the tightening and flattening of loose and bulging tissues above the fold, not only rejuvenating previously folded skin but also alleviating the degree of fold appearance during smiling or facial expressions [20,21]. Additionally, post-treatment collagen regeneration enhances skin firmness, thereby reducing the prominence of lines even when the face is at rest.

## 3. Cog Thread Implementation in the Reverse Technique

In the process of manipulating stretched tissues, such as the skin and nasolabial fat above the nasolabial fold line, careful consideration is essential for determining the procedural vector. This vector signifies the optimal direction in which the target tissue should be moved to achieve the most effective tightening of lax tissues. In the context of cog thread lifting for nasolabial folds, it is recommended to align the vector perpendicular to the fold line, as elucidated earlier. The procedure’s efficacy can be enhanced by maximizing tissue traction, pulling as much lax tissue as possible, even with the utilization of the same cog threads. One specific technique involves the insertion of the needle or cannula from the outer to inner and downward direction on the face. However, effectively engaging the lax skin and connective tissues adjacent to the nasolabial fold line with cog threads through this vector presents a notable challenge, particularly given the facial anatomical structure, a challenge that is particularly pronounced among Asian individuals [22,23,24].

The antegrade insertion of I-type cog threads from the outer to inner direction proves insufficient in addressing lax tissues around the nose and mouth. Conventional approaches typically involve utilizing the firm tissue around the hairline in the temporal region or the anthelix between the temple and ear as fixing points, with needles or cannulas inserted inward towards the nasolabial fold. However, the angular facial structure, particularly the prominent malar eminence area that protrudes when navigating towards the anterior aspect, necessitates the bending or angling downward of needles or cannulas to surpass this region. In the context of Korean facial anatomy, a CT scan frequently reveals an angularity between the medial and lateral cheeks, approximately 70 to 80 degrees, requiring an angled approach for needle or cannula insertion to navigate between these facial compartments (Figure 5).

Moreover, the use of steel needles or cannulas faces limitations in their ability to be easily bent beyond a certain extent to match the curvature of this region [26]. Changing the direction of advancement after raising the end of the needle or cannula upwards is also challenging, particularly in the area near the ears. The lateral face near the ears presents tough and firm skin and connective tissues, a thick SMAS layer, and a sparse deep fat layer beneath the SMAS. This results in an overall flattened adherence of the skin and tissues compared to other facial regions, making it practically challenging to pass needles or cannulas through and subsequently alter their direction upwards to face downwards, considering the strength of the tissues that need to be lifted (Figure 6).

If the insertion plane of the cannula around the protruding malar eminence fails to maintain a consistent depth due to bending, it may result in a shallow procedural plane. Conversely, attempting to prevent a decrease in the depth of cannula advancement by passing too deeply through the area where the face folds toward the nasolabial fold could inadvertently intersect with the origin areas of the zygomatic muscles near the malar eminence [27,28]. This interference might impede the lifting motion towards the mouth corners and, in individuals with developed malar eminence, could potentially accentuate the prominence of this region by collecting tissues toward the malar eminence [20,21,28]. In such cases, the Reverse Technique involves creating the entry point of the cog thread in the loose nasolabial fat area outside the nasolabial fold. Using the non-operating hand to grasp the subcutaneous fat tissue as thick as possible enables lifting before inserting the cannula. This allows the cannula to sufficiently pass through the target tissue, which is the nasolabial fat. After the cannula enters the subcutaneous fat layer, it follows along the SMAS layer, passing through the curved area of the face while maintaining a direction perpendicular to the nasolabial fold and toward the temporal and auricular regions. By lifting the inner nasolabial fat and guiding the cannula to pass beneath the subcutaneous fat layer toward the outer side of the face, the depth can be adjusted to pass through the SMAS layer or its vicinity in the lateral part of the face. Then, using the firm tissues present around the temple and ear as fixing points, the tip of the cannula can be pushed towards these areas to ensure that the cogs of the thread catch onto these firm tissues (Figure 7).

The Reverse Technique involves creating the entry point of the cog thread below and inserting the thread from below to above. This technique offers several advantages, including sufficient engagement with the lax tissues in the target area, consistent depth of insertion and progression of the cog thread in folded regions of the face, and the ability to utilize the thread to its maximum length. In contrast to the conventional method, where the I-type cog thread is commonly inserted from top to bottom, the Reverse Technique ensures that tissues above the insertion point are not inadvertently pulled downwards. Regardless of the structure and shape of the I-type cog threads, inserting the thread from bottom to top leads to a direction of pulling force from lax tissues toward firm tissues when traversing both types of tissues simultaneously. The force applied by pulling after the cog thread has engaged both lax and firm tissues tends to focus in one direction. Regardless of the orientation of the threads holding the tissue, the directional force prioritizes a direction other than the orientation of the threads. The direction of the threads inserted by the cog threads can be divided into forward and reverse directions. Tissues located in the forward direction of the threads are not necessarily moved in the opposite direction. For example, inserting a bidirectional cogged thread (I-type) in reverse, from below the face to above, places the forward-oriented threads towards the cranial side and the reverse-oriented threads towards the inferior facial area. In this scenario, the tissues located in the area of the face where the reverse-oriented threads are situated are pulled towards the tissues of the cranial side where the forward-oriented threads are positioned, rather than the upper tissues being pulled downwards.

The conceptual basis of this phenomenon bears analogy to commonplace scenarios, envisioning a tug-of-war game serves as an illustrative example. In this context, participants hold onto a rope wrapped around the waist while encountering significant resistance from an opponent refusing to pull back. Attempting to exert a strong force towards oneself does not result in the opponent moving; instead, the opposing force prompts the individual to move in the opposite direction, akin to the physical force exerted in this scenario. Consequently, for optimal efficacy in thread lifting, it becomes imperative not only to skillfully pull and engage loose and lax tissues using threads but also to identify fixing points consisting of firm tissues resistant to easy displacement. Engaging these firm tissues is crucial in preventing the threads, which secure the loose tissues, from slipping downward, thereby enhancing the effects of thread lifting.

To judiciously utilize fixing points for placing cog threads that secure slackened tissues, one must consider the direction in which the target tissue needs to be pulled most efficiently to stretch the slackened tissue. Deliberating on the movement direction becomes pivotal, determining how differently the threads should be inserted to form vectors for attachment to the fixing points. When contemplating the implementation of vertical and oblique direction cog thread lifting for specific facial areas, the depiction of the insertion sites and trajectory of traditional method cog threads is illustrated in Figure 8.

If the Reverse Technique with I-type cog thread lifting is applied across the entire face, the insertion sites and trajectory of the threads undergo modifications, as depicted in Figure 9.

When employing the Reverse Technique with cog threads to elevate lax tissues situated outside the nasolabial folds upwards, intending to enhance the overall appearance around the sides of the nose and mouth and specifically addressing the boundaries of residual wrinkles, it is essential to initially consider the differing thicknesses or firmness of the tissues above and below the nasolabial folds and around the central wrinkles, as illustrated in Figure 10.

## 4. Utilizing Volumizing Threads in the Cross Technique

The conventional method traditionally involved the use of fillers or mono threads inserted along existing wrinkle lines to mitigate creases. In contrast, the author advocates for a distinct approach termed the “Cross Technique using volume-oriented multi-thread PDO sutures” when targeting the smoothening of the boundary between the firm tissue beneath the wrinkle line and the folded line above, ultimately reducing its depth. In this Cross Technique, the entry point, slightly inside the nasolabial fold, is strategically determined, considering the length of the volume thread relative to how far beneath the wrinkle line it needs to reach. Subsequent to needle puncturing, the suture is inserted through the thick subcutaneous tissue firmly attached to the SMAS, perpendicular to the wrinkle line, from bottom to top. As the needle crosses the wrinkle line, its end traverses under the retinaculum cutis in the dermis beneath the loose tissue above the wrinkle line, utilizing the volumizing section of the thread within the thick tissue beneath the wrinkle line as a support structure. The upper segment of the suture exerts pressure against and flattens the loose tissue above the wrinkle line, rectifying the discrepancy in skin layer height resulting from the differing firmness between the tissues above and below the wrinkle line. By strategically placing multiple volume threads in the vertical direction of the wrinkle line, collagen growth follows the thread’s orientation, effectively tightening the loose tissue above the wrinkle line. Consequently, this procedure mitigates the folding and compression of the skin around the nose and mouth induced by facial muscle movements (Figure 11).

Inserting volume-oriented threads into firm tissues to serve as a supportive framework for pressing and flattening loose tissues represents one approach. However, in cases where the objective is solely to rejuvenate sagging areas, as opposed to utilizing them as a support structure, volume threads can be placed along the wrinkle lines. These threads, integrated within the skin, promote enhanced collagen regeneration compared to standard PDO mono threads, yielding a noteworthy effect in revitalizing and firming the skin.

## 5. Anatomical Considerations

During the procedural intervention, careful consideration must be given to vascular structures in specific regions. In the nasolabial fold area, the facial artery, ascending along the wrinkle lines, necessitates attention. In the mid-cheek area, critical landmarks include the premasseteric branch of the facial artery, tracing the anterior border of the masseter muscle, and the transverse facial artery, ascending subcutaneously beneath the zygomatic arch from the superficial temporal artery [29,30,31,32,33]. Furthermore, in the lateral face region, diligence is required to prevent injury to the ascending superficial temporal artery along the preauricular crease (Figure 12).

The facial artery typically courses through both the upper and lower areas of this region, aligning with the direction of facial expression muscles, such as those in the nasolabial fold, during procedures involving cog threads or volumizing agents. Diligent manipulation of the cannula during its movement helps minimize the risk of injury. Conversely, a higher risk of injury is associated with needle use in monofilament procedures aimed at reducing the fold, compared to the cannula. In the application of the Reverse Technique with cog threads, the entry point is established in the outer nasolabial fat region, adjacent to the nasolabial fold. Anatomically, in over 70% of Koreans, the facial artery is observed to cross the nasolabial fold. Even when positioned externally, more than half of these vascular structures pass within 5mm of the fold’s edge (Figure 13) [34,35].

Therefore, by creating an entry point a few millimeters outside the actual crease and inserting the cannula tip accordingly, the likelihood of vascular injury can be significantly reduced. Moreover, as the nasolabial fat extends obliquely along the outer region of the nasolabial crease, there is no essential need to establish the entry point for cog threads inside the crease itself to address the fatty tissue protrusion above the nasolabial fold (Figure 14).

When executing the cross technique using volumizing threads, creating an entry point inside the nasolabial fold line is necessary for thread insertion, passing through the thick subcutaneous tissue above the orbicularis oris muscle. However, in over half of the Korean population, facial arteries, including the superior labial artery branching off from the facial artery, course superficially over the orbicularis oris muscle, towards the nasolabial fold line [36]. This superficial exposure of the facial arteries increases the concern for inadvertent vascular injury during needle puncture or thread insertion [37]. Given that facial arteries often run close to the nasolabial fold, creating an entry point several millimeters away from the fold can potentially prevent direct needle penetration into the vessels and consequent bleeding.

When inserting the cannula horizontally along the wrinkle lines, maintaining a smooth and uniform depth is crucial to minimize the risk of vascular injury. Additionally, this approach aims to ensure a smoother skin surface after the volumizing thread procedure, reducing the likelihood of irregularities or unevenness in the skin texture.

During the reverse technique using barbed threads, when inserting the cannula through the entry point to pass the tip below the nasolabial fat, it is imperative to engage the ligamentous structures, serving as hanging points. These ligaments should be attached to the threads to enable them to interact with the hanging tissues linked to the adjacent fixing points near the ear [38,39]. Meanwhile, the zygomatic ligament, originating from the bone, presents a more robust ligamentous tissue, pivotal during barbed thread lifting, as its tension assists in alleviating the prominence of the nasolabial fold by pulling it towards the side of the nostril and effectively improving its appearance (Figure 15) [40,41,42,43].

As the cannula traverses the boundary area between the medial and lateral face, altering the treatment plane slightly deeper along the lateral face by perforating the SMAS layer and progressing below it allows for the advancement of the barbed threads without concern for potential damage to the superficial temporal artery and premasseteric branch. When advancing the barbed threads perpendicular to the nasolabial fold line on the cheek, the fixing point is commonly located at the midpoint of the ear, often utilizing the robust fascial tissue known as Lore’s fascia situated just in front of the tragus as the fixing point [39]. Typically, as the cannula crosses the lateral face and approaches the SMAS layer beneath, an encounter with solid resistance is felt toward the tragus, signifying Lore’s fascia [39,44,45]. This structure extends from the ear cartilage through deep and superficial muscular layers, connecting robust fibers resembling ligaments with the skin. An anatomical view reveals the rectangular-shaped, sturdy ligamentous tissue connecting the parotid gland and the ear cartilage below the tragus when the parotid fascia is displaced anteriorly (Figure 16).

## 6. Technical Guideline

As previously discussed, meticulous care is imperative when employing the reverse technique with barbed threads, emphasizing the need to guide the threads towards the outer region of the nasolabial fat, outside the nasolabial fold, and towards the tragus. The thin SMAS layer near the nasolabial fold may give a less rigid sensation, and carelessness in proceeding into excessively loose areas could lead to deep penetration into the deep fat layer. This may potentially irritate the inner mucous membrane of the mouth, causing discomfort akin to the presence of foreign matter upon mouth opening. Therefore, caution must be exercised.

Execution of the procedure involves gripping the loose skin and connective tissue of the target area, i.e., the loose and sagging nasolabial fat, with the left hand, lifting it upward substantially. Subsequently, inserting the cannula tip from the underside to the upper side through the fat pad in a manner reminiscent of sewing with a needle is sufficient.

Post-procedure, as the nasolabial fat above the previously protruding crease flattens and moves outward, there is an apparent reduction in volume, resulting in a flattened appearance. This decrease in height disparity between the side of the nose and the adjacent cheek softens the previously stepped boundary, reducing the prominence of the nasolabial fold. Furthermore, as tissues move towards the mid-cheek groove, an area that tends to appear relatively diminished, an overall smoothing effect is observed in the entire mid-cheek region.

When the patient is lying down and slightly elevates the chin, one can observe the sagging tissues above the nasolabial fold shifting towards the outer eye area. In the supine position with a slight elevation of the chin, sagging tissues above the nasolabial fold shift towards the outer eye area. During this posture, the insertion of barbed threads through the entry points is conducted. For an optimal lifting effect, variously directed I-type multidirectional cogged threads can be used, each producing distinct tensile strength [22,46,47,48,49,50]. However, for emphasizing tensile strength, the preference is for a specialized bidirectional cogged thread designed for reverse procedures. This specialized thread, featuring more barbs towards the handle, facilitates increased entanglement with looser tissues when inserted from bottom to top. It is essential to note that the end portion of the thread contains fewer barbs; thus, effective identification and securement of these less barbed areas to solid fixing points are crucial. This ensures the maintained integrity of the pulled loose tissues and prevents subsequent sagging.

The thickness and quantity of specialized bidirectional cogged threads designed for reverse procedures are tailored based on factors such as skin thickness, elasticity, the degree of skin and tissue laxity, and overall volume. During the insertion of these reverse-specific threads, their progression from the inner to the outer side should be carefully adjusted. As previously elucidated, a change in the insertion layer of the barbed threads is necessary at the border between the medial and lateral areas of the face. Typically, transitioning the insertion plane of the threads from the deep subcutaneous fat layer to beneath the SMAS layer while moving outward from the vertical line drawn from the lateral orbital rim accomplishes this alteration (Figure 17).

In some instances, a band-like bulge may manifest at the starting point of the nasolabial fold, where the tip of the nose meets the cheek, creating a prominent protrusion. To address this, solely pulling the midsection of the nasolabial fold might not effectively tighten the lax skin and connective tissue at this starting point. Therefore, when targeting the area adjacent to the nasal tip, where Koreans often have a prominence known as the malar eminence, altering the direction of the barbed thread’s path for pulling becomes crucial. Utilizing the lateral orbital thickening of the orbital rim near the eye socket as a fixing point is advisable in cases where creating an entry point at the temple hairline and traversing the temple area might be challenging. Effective treatment can be achieved by creating an entry point on the band of lax fatty tissue near the nasal tip, inserting a short reverse-specific bidirectional cogged thread, guiding it toward the eye area without traversing the malar eminence, and then securing it properly to prevent subsequent loosening at the lateral orbital thickening (Figure 18).

Post-procedurally, there is evident tissue tightening, leading to enhancements in contour changes attributed to regional volumetric variations. Even subsequent to interventions involving barbed threads, residual marks may persist where tissue folding has occurred along the demarcation between the cheek and the nasolabial region. To mitigate such imprints, the application of the cross technique utilizing volume threads, as elucidated earlier, serves to refine these folds while imparting resilience to the skin.

In cases where the objective is the refinement of crease lines without substantial sagging in the nasal alar region, the consideration of inserting volume threads along the marginally inner expanse of the crease lines is warranted. Determination of the entry point for thread insertion involves a thoughtful assessment of the required thread length just beneath the lower terminus of the crease lines. Subsequent to needle puncture, the insertion of the thread into the subcutaneous tissue above the orbicularis oris muscle, coupled with the adherence to the crease lines by bottom-to-top thread insertion, can effectively achieve the intended outcome.

## 7. Discussion

The administration of fillers to address nasolabial folds is a common practice in cosmetic procedures. However, inadvertent injection into the facial artery poses severe complications, including skin necrosis and irreversible blindness [51]. In the event of blindness, irreversibility is a distinct consequence, while skin necrosis may lead to the formation of permanent scar tissue. Consequently, thread injections have emerged as a novel approach for addressing deep nasolabial folds. Despite their tendency to avoid severe complications seen with fillers, it is imperative to consider anatomical characteristics when correcting deep nasolabial folds using threads. The facial artery, which courses along the nasolabial fold both laterally and medially and intersects the fold, must be preserved as a critical consideration in thread-based correction procedures [35].

In summary, nasolabial folds exhibit distinct types based on causative factors. Type one involves volume deficiency beside the nasal alar, resulting in hollowing in the buccal area and the formation of deep lines, a condition for which fillers have replaced older methods like implants. Type two entails sagging of the anterior buccal fat pad, creating a contrast in tissue thickness above and below the wrinkle line. The third type involves muscle fibers pulling the skin during smiling, deepening the fold. Thread-lifting methods are designed to target specific tissue layers, thereby reducing fold depth and enhancing appearance. Often, a combination of these types contributes to the appearance of the fold. Interventions aim to address volume discrepancies and muscle attachments, reducing fold depth and improving skin appearance during facial expressions, with collagen regeneration playing a role in post-treatment skin firmness.

This guideline delineates anatomical considerations pertinent to cosmetic procedures in the facial region, emphasizing the significance of comprehending vascular structures, particularly within the nasolabial fold, mid-cheek, and lateral face areas. It underscores potential risks to facial arteries during interventions involving cog threads or volumizing agents.

In the context of the nasolabial fold, meticulous care is warranted due to the presence of the facial artery following the wrinkle lines. The guideline recommends establishing an entry point slightly outside the fold to mitigate the risk of vascular injury during procedures. For volumizing thread techniques, it underscores the necessity of creating an entry point inside the fold. However, given the superficial exposure of facial arteries over the orbicularis oris muscle, it advocates establishing the entry point several millimeters away from the fold to prevent potential vessel injury.

During procedures involving barbed threads, the engagement of ligamentous structures is deemed critical for effective tissue lifting and relocation. Ligaments such as the outer maxillary and zygomatic ligaments play a pivotal role in pulling and maintaining tissue at the relocated position. Moreover, adjusting the treatment plane slightly deeper along the lateral face, especially when encountering the SMAS layer, facilitates the safer advancement of barbed threads without risking damage to vascular structures.

Fixing points for barbed threads are frequently located at the midpoint of the ear, utilizing robust fascial tissues like Lore’s fascia, which extends from the ear cartilage through muscular layers, connecting ligamentous fibers with the skin. These anatomical landmarks and structures serve as indispensable reference points for the safe and effective execution of thread-based cosmetic procedures in the facial area.

This technical guideline centers on the “Reverse Technique,” employing barbed threads for lifting and tightening sagging tissues in the nasolabial fold area. It underscores caution when inserting threads towards the loose fat outside the fold, as excessive penetration could potentially irritate the inner mouth mucous membrane. The procedure involves gripping the loose skin and connective tissue, lifting it upwards, and inserting the cannula tip through the fat pad. The outcome encompasses a reduction in the prominence of the nasolabial fold, a decrease in the disparity between the nose and cheek, and a smoothing effect on the mid-cheek region.

When the patient assumes a supine position with a slightly elevated chin, barbed threads are introduced through entry points to lift sagging tissues above the fold towards the outer eye area. The technique recommends the use of specialized bidirectional cogged threads for enhancing tensile strength, ensuring effective entanglement with loose tissues and preventing post-procedure sagging. The selection of thread thickness and quantity is contingent on various factors, including skin thickness, elasticity, laxity, and volume.

Furthermore, the guideline underscores the alteration of the thread insertion path to address specific areas, such as the starting point of the nasolabial fold near the nose tip or the malar eminence area. It suggests leveraging lateral orbital thickening near the eye socket as a fixing point to tighten the fatty layer, offering effective treatment without traversing sensitive areas. The procedure yields noticeable tightening of lax tissues, reshaping due to regional volume differences, and potential marks at the border between the cheek and nasolabial area. To refine these marks, the guideline recommends employing the “Cross Technique” using volume threads, which can refine folds and add firmness to the skin. Additionally, for enhancing crease lines without significant sagging in the nasal alar area, the insertion of volume threads along the crease lines is considered for achieving the desired effect.

## Figures and Tables

**Figure 1 diagnostics-14-00716-f001:**
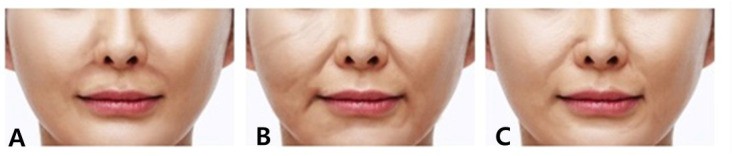
The visual classification of nasolabial folds into three distinct types: Type 1 (**A**) delineates a depression in the paranasal region, Type 2 (**B**) signifies a sagging or drooping of the nasolabial fat, while Type 3 (**C**) illustrates a crease formed due to the action of upper lip elevator muscles.

**Figure 2 diagnostics-14-00716-f002:**
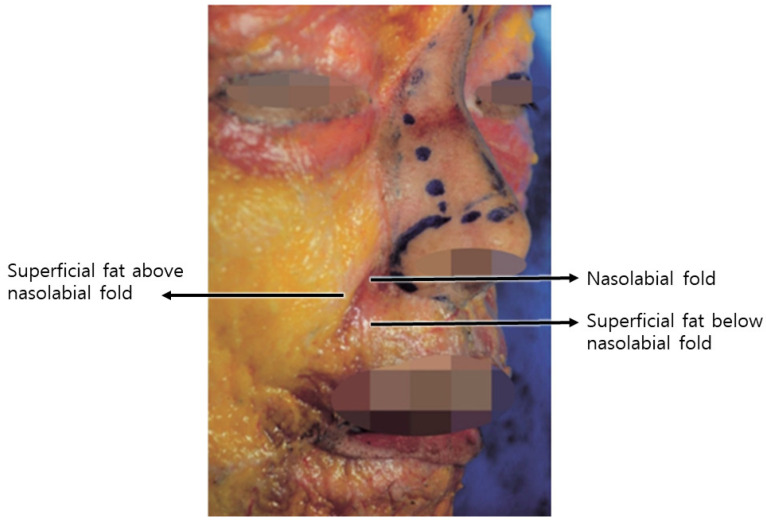
Visually represents the differences in tissue thickness surrounding the nasolabial folds.

**Figure 3 diagnostics-14-00716-f003:**
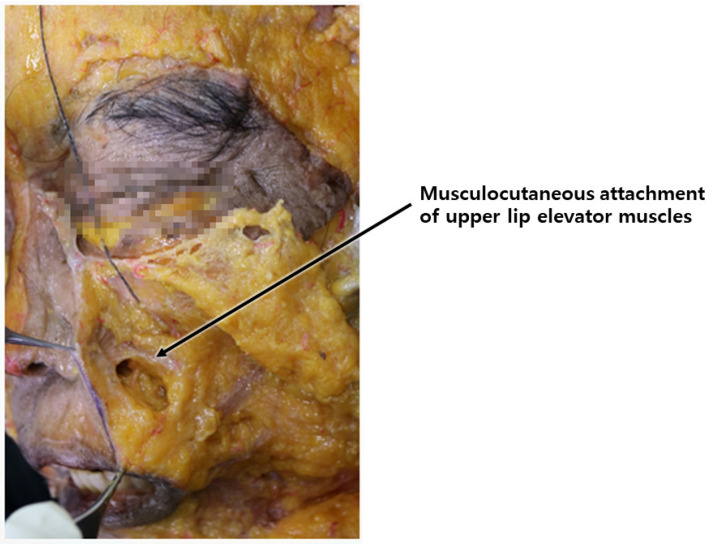
Illustrates the musculocutaneous attachment points of upper lip elevator muscles beneath the nasolabial folds, emphasizing their anatomical connections.

**Figure 4 diagnostics-14-00716-f004:**
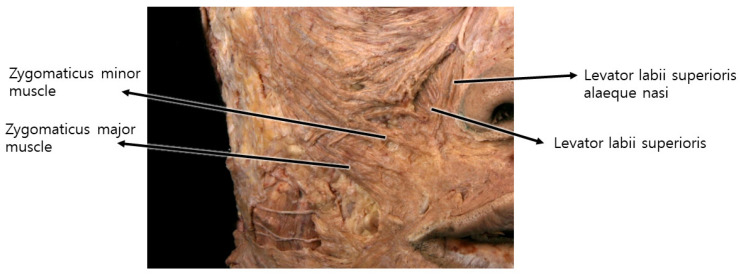
Represents the upper lip elevator muscles in relation to the nasolabial fold, likely showcasing their positioning and influence on facial expressions in this area.

**Figure 5 diagnostics-14-00716-f005:**
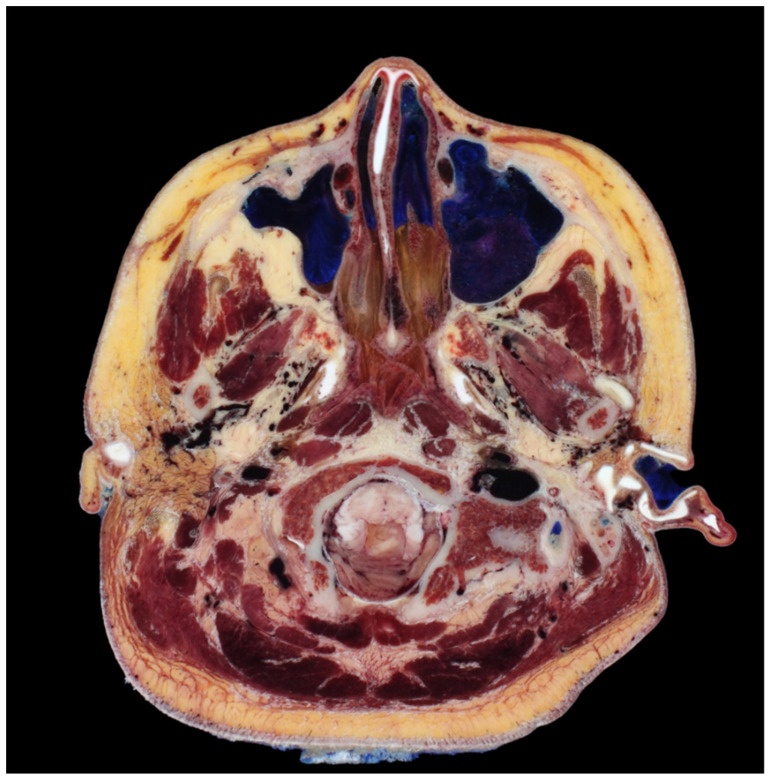
A transverse section revealing the degree of curvature or bending between the medial and lateral cheek regions, providing insights into their structural differences. Cross-sectional head images from (Park et al., 2005) [25].

**Figure 6 diagnostics-14-00716-f006:**
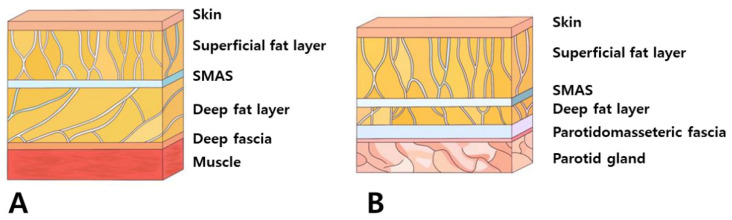
Highlights variations in the organization of subcutaneous tissue between the fundamental facial area (**A**) and the parotid region (**B**), potentially indicating differences in tissue composition or structure.

**Figure 7 diagnostics-14-00716-f007:**
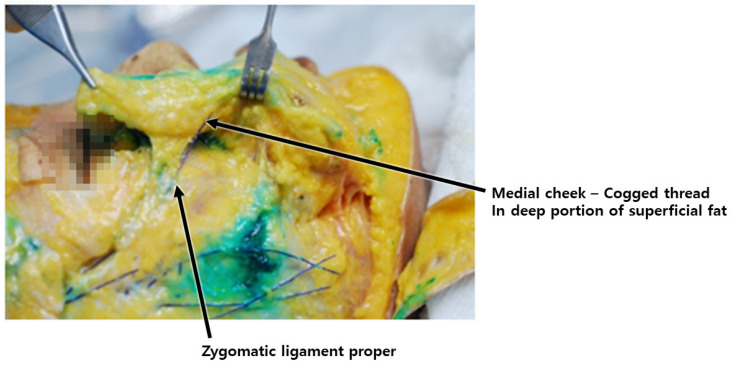
Shows alterations in the insertion plane of cogged threads between the medial and lateral cheek regions, demonstrating potential differences in thread placement or direction for facial lifting. Threads inserted are Bluerose Forte L (Hugel, Inc., Chuncheon, Republic of Korea), Licellvi X (JWorld Co., Ltd., Tokyo, Japan), and countourel ultra (Croma Pharma GmbH, Leobendorf, Austria).

**Figure 8 diagnostics-14-00716-f008:**
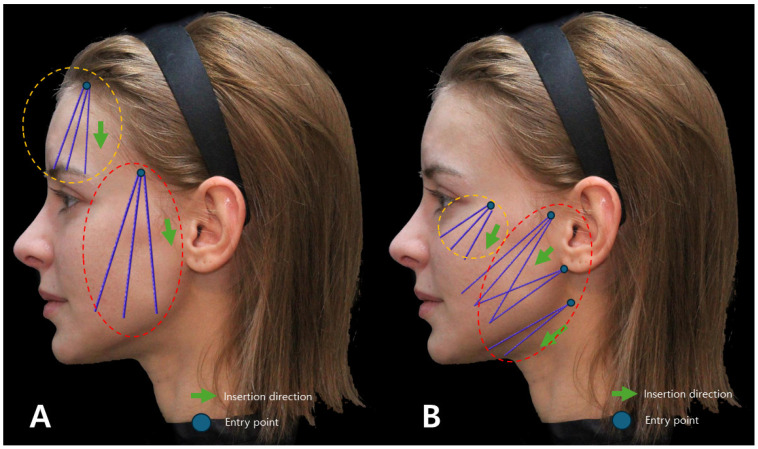
Depicts the conventional design (**A**) and direction of thread lifting using I-type cogged threads, likely showcasing the standard approach for this specific thread type (**B**). Thread in orange dotted are Bluerose Forte L (Hugel, Inc.), Licellvi X (JWorld Co., Ltd.), and countourel ultra (Croma Pharma GmbH). The threads in red dotted are Bluerose Forte L/F (Hugel, Inc.), Licellvi X/FIX (JWorld Co., Ltd.), countourel ultra/ultra+ (Croma Pharma GmbH).

**Figure 9 diagnostics-14-00716-f009:**
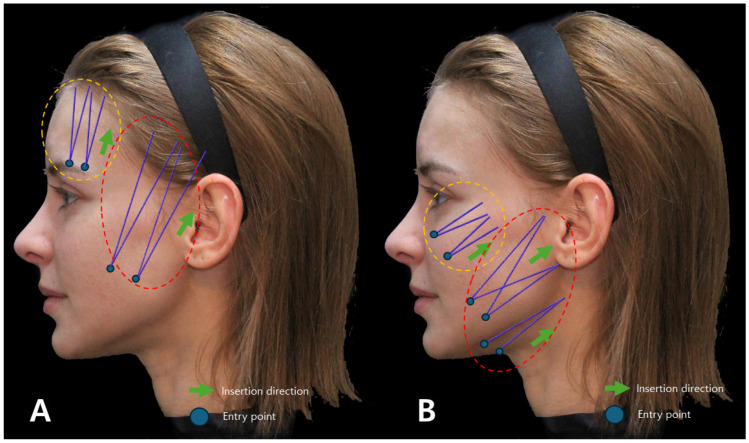
Represents the design and direction of I-type cogged thread lifting for the Reverse Technique (**A**), potentially showing an alternative or modified approach in thread application for specific aesthetic outcomes (**B**). Thread in orange dotted are Bluerose Forte L (Hugel, Inc.), Licellvi X (JWorld Co., Ltd.), countourel ultra (Croma Pharma GmbH). The threads in red dotted are Bluerose Forte L/F (Hugel, Inc.), Licellvi X/FIX (JWorld Co., Ltd.), and countourel ultra/ultra+ (Croma Pharma GmbH).

**Figure 10 diagnostics-14-00716-f010:**
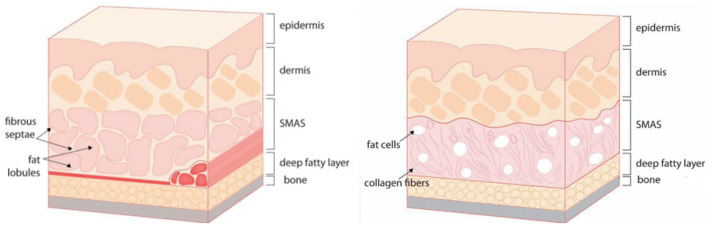
Displays the structural variances between layers covering and not covering the bones around the nostrils and mouth, possibly indicating differences in tissue composition or thickness.

**Figure 11 diagnostics-14-00716-f011:**
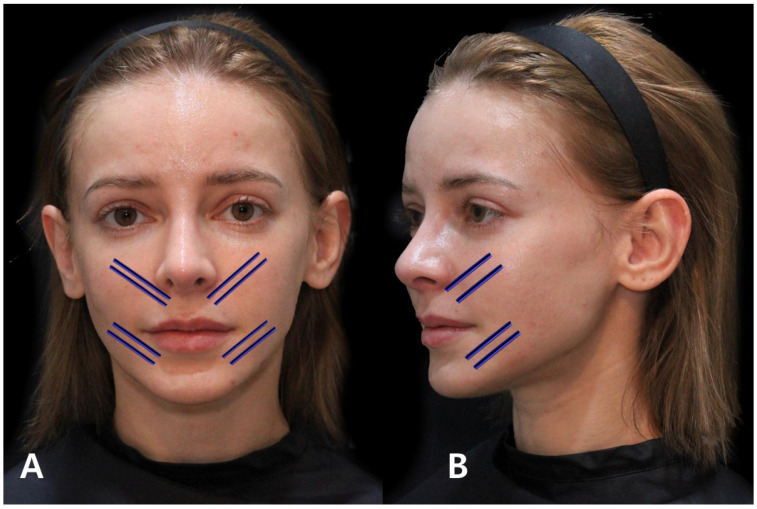
Demonstrates the entry point and direction of the Cross Technique using volumizing PDO threads for addressing nasolabial folds and marionette lines, offering insights into specific thread insertion methods for targeted facial areas. The thread used are Licellvi Jamber F (JWorld Co., Ltd.) and countourel Jamber F (Croma Pharma GmbH). Frontal view (**A**) and lateral view (**B**).

**Figure 12 diagnostics-14-00716-f012:**
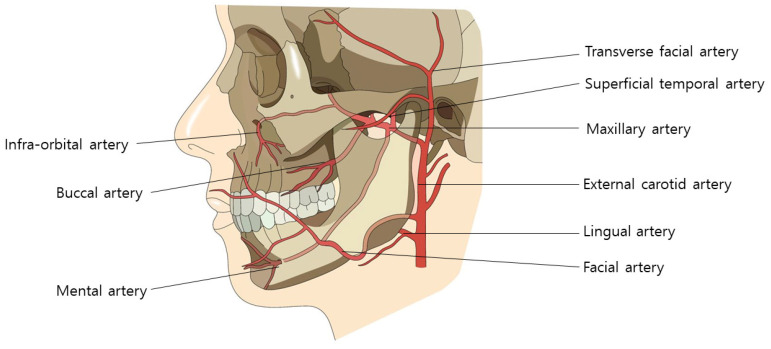
Arteries in mid-facial cheek area: illustrates the arterial network in the mid-facial cheek, aiding in understanding blood supply for surgical or minimally invasive procedures.

**Figure 13 diagnostics-14-00716-f013:**
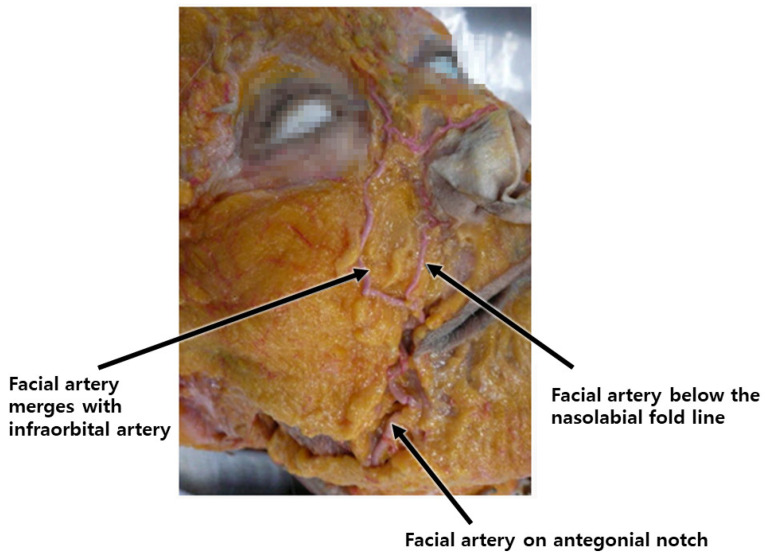
Position of facial artery around nasolabial fold: shows the location and course of the facial artery concerning the nasolabial fold, crucial for procedures in this area.

**Figure 14 diagnostics-14-00716-f014:**
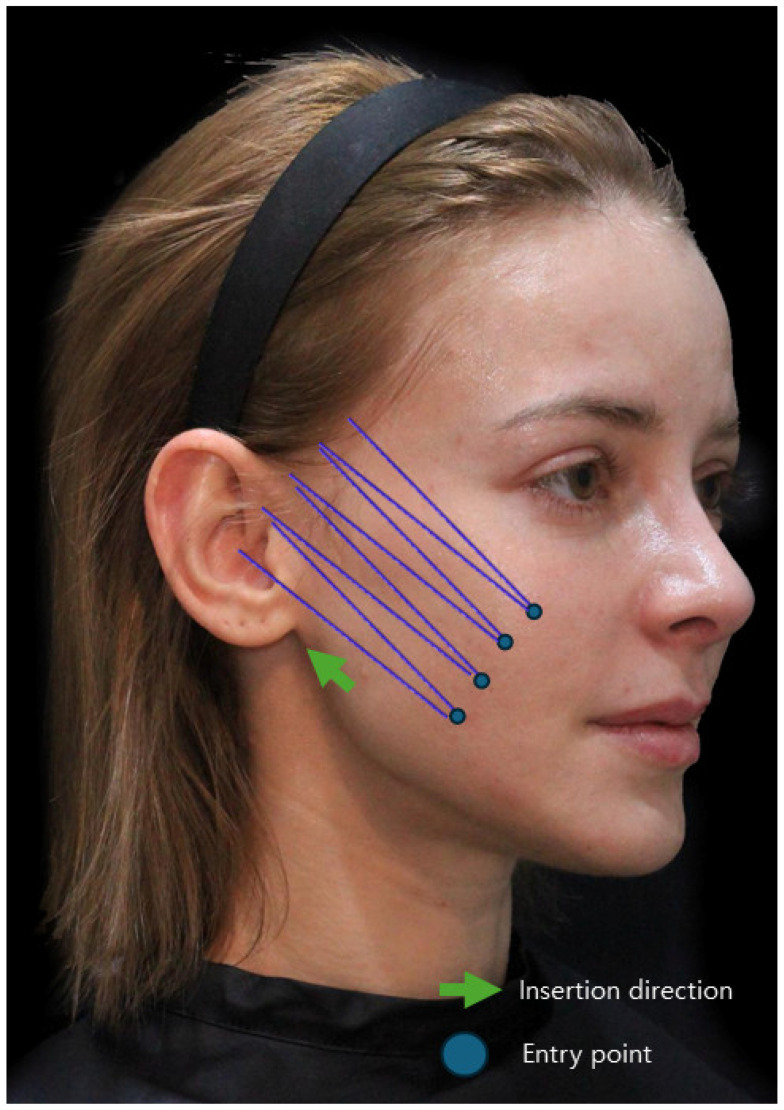
Entry point and direction for reverse technique using I-type cogged threads: Displays insertion points and 367 directions for I-type cogged threads in the Reverse Technique for nasolabial folds and marionette lines. The thread used are Bluerose Forte L/F (Hugel, Inc.), Licellvi X/FIX (JWorld Co., Ltd.), and countourel ultra/ultra+ (Croma Pharma GmbH).

**Figure 15 diagnostics-14-00716-f015:**
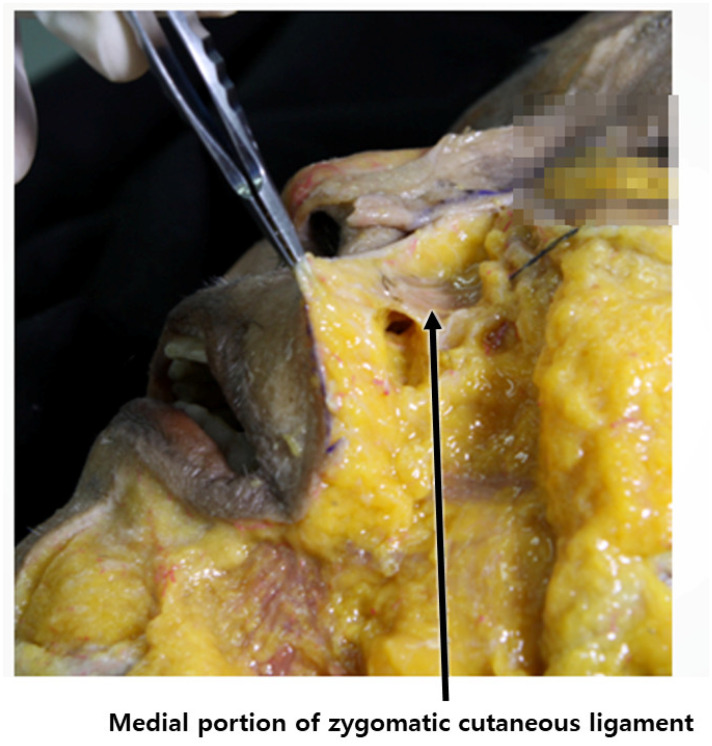
Location of medial portion of zygomatic cutaneous ligament: depicts the specific position of the maxillary ligament within facial anatomy.

**Figure 16 diagnostics-14-00716-f016:**
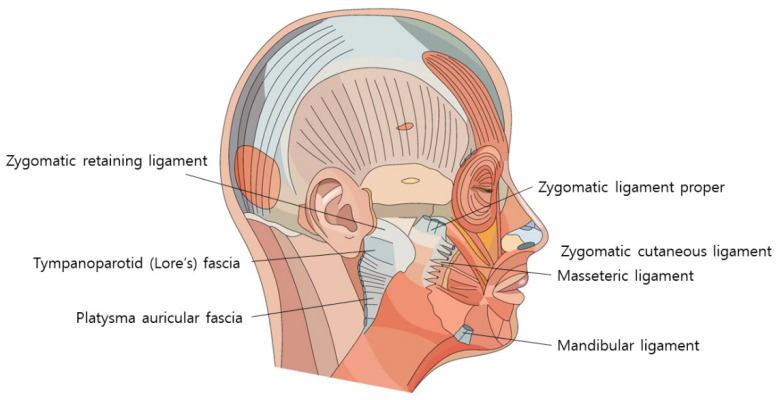
Location of Lore’s fascia: illustrates the exact positioning of Lore’s fascia, important for surgical procedures targeting facial rejuvenation.

**Figure 17 diagnostics-14-00716-f017:**
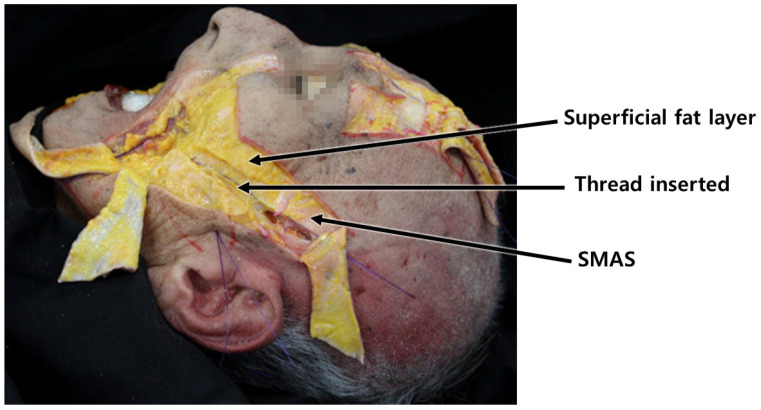
Difference in insertion plane of cogged threads between medial and lateral cheek region: Demonstrates variations in thread insertion direction between cheek regions, aiding in treatment planning. Threads inserted are Bluerose Forte L (Hugel, Inc.), Licellvi X (JWorld Co., Ltd.), and countourel ultra (Croma Pharma GmbH).

**Figure 18 diagnostics-14-00716-f018:**
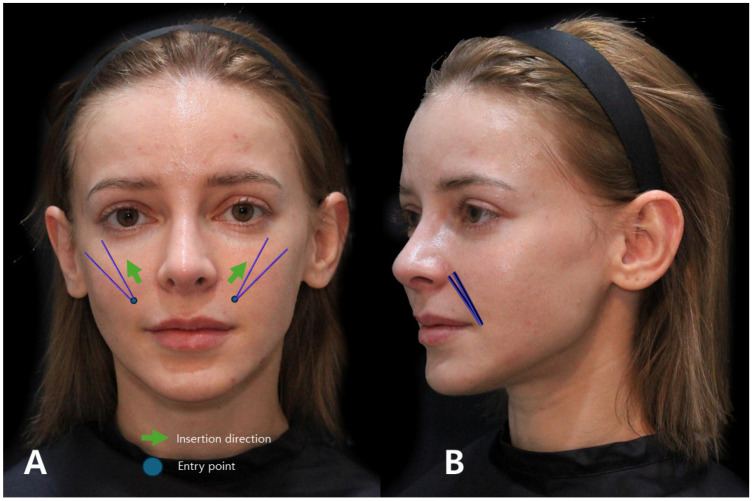
Designs for cogged and volumizing thread insertion in nasolabial area: displays tailored thread designs for addressing specific concerns like nasolabial fat and crease, assisting in customized treatments. The panel (**A**) is nasolabial fold treatment with reverse techinique for correction of nasolabial fold Bluerose Forte L (Hugel, Inc.), Licellvi X (JWorld Co., Ltd.), countourel ultra (Croma Pharma GmbH). Panel (**B**) is correcting it with spring type thread (Licellvi Jamber F (JWorld Co., Ltd.), countourel Jamber F (Croma Pharma GmbH)).

## Data Availability

Not applicable.

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
