# Peer review of "Why Do Nasolabial Folds Appear? Exploring the Anatomical Perspectives and the Role of Thread-Based Interventions"

_diagnostics, 2024, doi:10.3390/diagnostics14070716_

Round 1

Reviewer 1 Report

Comments and Suggestions for Authors

Thank you for your submission which gives an extensive overview of nasolabial folds and how to address them using barbed threads.

You describe 3 types of nasolabial folds but then also state that most are a combination of 2 or 3 of these. Would it not be clearer to describe 3 main causes of nasolabial folds and then classify according to the cause?

Figure 6 is a colour image, labeled as a CT scan. Is it a CT that has been colourised or is it a cadaver?

How is the entry point for the thread created? 

Author Response

Dear. Reviewer 1 

You describe 3 types of nasolabial folds but then also state that most are a combination of 2 or 3 of these. Would it not be clearer to describe 3 main causes of nasolabial folds and then classify according to the cause?
- We have rearranged and corrected as accordingly. Thank you so much. 

Figure 6 is a colour image, labeled as a CT scan. Is it a CT that has been colourised or is it a cadaver? 
- The legend for Figure 6 has been corrected. Thank you so much for noticing.

How is the entry point for the thread create?
- Most of the thread their entry point are made by needle in advance to the thread procedures.  

Thank you so much for reviewing the paper. 

Reviewer 2 Report

Comments and Suggestions for Authors

The authors did a narrative review on the issue of nasolabial fold and propose a technique based on anatomical consideration of the area. In general, the review is well written. Few concerns or suggestion in my view are:

1.       In the abstract there are too many sentences starting with the word “this review”. Kindly review and make changes.

2.       Subsection 2 title is “The Pathogenesis of Nasolabial Folds”. Why used the term pathogenesis? It is like nasolabial fold is a pathology. Would suggest replacing the term.

3.       The pictures (clinical and diagram) with permission? If the cadaver are authors original pictures, can they blank the eyes to protect the identity. For the cadaveric dissection (if the pictures are authors own), any ethical approval obtained?

4.       The referencing of the review is rather odd and difficult to follow.  Example paragraph line 131 to 138 and in paragraph 163 to 174, the citation number is at the end of the sentence. How can readers identify which facts relates to which citation? It is an odd way to cite. Kindly cite the facts at the end of the sentence of the said facts for easy referencing.

Comments on the Quality of English Language

Not much issue, some repetition of similar words but otherwise OK

Author Response

Dear. Reviewer 2 

  1.       In the abstract there are too many sentences starting with the word “this review”. Kindly review and make changes.- Thank you so much for the comment. Abstract has been amended.

2.       Subsection 2 title is “The Pathogenesis of Nasolabial Folds”. Why used the term pathogenesis? It is like nasolabial fold is a pathology. Would suggest replacing the term. -

All of them are corrected. We have replaced pathogenesis with etiology.

3.       The pictures (clinical and diagram) with permission? If the cadaver are authors original pictures, can they blank the eyes to protect the identity. For the cadaveric dissection (if the pictures are authors own), any ethical approval obtained? - -
- We have covered all the eye with the mosaic. All of korean  cadavers are from institute which is all donated and are approved for research purposes. There will be no issues.

4.       The referencing of the review is rather odd and difficult to follow.  Example paragraph line 131 to 138 and in paragraph 163 to 174, the citation number is at the end of the sentence. How can readers identify which facts relates to which citation? It is an odd way to cite. Kindly cite the facts at the end of the sentence of the said facts for easy referencing. 
-

In paragraphs 131 to 138, the citations have been amended to provide clarity.  In paragraphs 163 to 174, the citations are positioned at the end of the sentence because the preceding sentence contains the fact being referenced.

Thank you so much for reviewing the paper.